# Pre-Harvest Benzothiadiazole Spraying Promotes the Cumulation of Phenolic Compounds in Grapes

**DOI:** 10.3390/foods11213345

**Published:** 2022-10-25

**Authors:** Yumei Jiang, Faisal Eudes Sam, Jixin Li, Yang Bi, Tengzhen Ma, Bo Zhang

**Affiliations:** Gansu Key Laboratory of Viticulture and Enology, Gansu Wine Industry Technology R&D Center, College of Food Science and Engineering, Gansu Agricultural University, Lanzhou 730070, China

**Keywords:** benzothiadiazole, phenolic compounds, elicitor, grape, quality

## Abstract

Benzothiadiazole (BTH) is a commercial chemical elicitor that can induce an innate immune response in grapevines and improve the phenolic components and color quality of grapes and corresponding products. The literature on the influence of BTH on the accumulation and metabolism of phenols from grapes is extensive. However, many unknown bio-mechanisms involved have been poorly investigated, which opens a gateway for pioneering research that needs to be done in this field. To this purpose, this review aims to analyze and explore the gaps in current research so that subsequent studies may be geared towards them.

## 1. Introduction

Grape is a valuable economic crop from which various products such as raisins, table grapes, grape juice, and wine (the most significant commercial product) can be obtained. Grape quality is greatly influenced by secondary metabolites such as phenolic compounds [1,2,3,4]. Agronomic techniques, including leaf removal [5,6], canopy systems [2], foliar fertilization [7], irrigation [8,9], application of exogenous substances [10,11,12,13], planting grass in the vine field [14], and bunch thinning [15] can affect the secondary metabolism of grapes. Some practices can directly influence the secondary metabolism of the berries. However, in some cases, primary metabolism is affected first, which may indirectly affect secondary metabolism [16].

Prevalent diseases such as gray mold (*Botrytis cinerea*), powdery mildew (*Erysiphe necator*), and downy mildew (*Plasmopara viticola*) impair the healthy development of grapevines and the quality of their berries [16,17]. Chemical pesticides are the most effective and common method of preventing or combating such diseases, but they or their degradation products negatively affect human health and the environment [16,18]. At present, green and climate-friendly production strongly calls for innovative and friendly disease control measures to ensure healthy and high-quality grape production. Therefore, studies on plant pathogen defense by elicitors/inducers in viticulture have attracted much attention [19]. Elicitors can stimulate an innate immune response in the plant [4,18,20], effectively reducing the incidence of plant diseases. Elicitors have the characteristics of safety, broad-spectrum, sustainability, and lower environmental hazards, toxicity, and ecological risks than pesticides and genetically modified plants [18]. They can be an effective alternative to conventional agrochemicals in defending against biotic and abiotic stresses [21,22,23,24]. Elicitors can cause various localized or systemic plant defensive reactions, such as an increase in oxidative burst and defense genes expression, changes in the cell wall composition [15], and the buildup of anti-microbial substances such as phenolic compounds [22].

The importance of phenolics in plants is beyond question as they help plants defend against biotic and abiotic stresses [25], enhance the organoleptic and nutritional properties of fruits and corresponding products [26,27,28], and have effects on human well-being [29,30]. Grapes are an abundant source of polyphenols [27,31,32,33], particularly flavanols, flavonols, anthocyanins, proanthocyanidins, hydroxycinnamic acids, and trace amounts of stilbenes [34,35,36]. Interest in grape phenolics is growing because of their potential to prevent various oxidative stress-related diseases and because of their cardioprotective [37], anticancer, anti-inflammation [38], anti-aging, and anti-microbial characteristics [31,32,34,39]. Moreover, grape phenolic compounds not only have positive effects on certain types of degenerative diseases [40] but are also essential in developing wines and other finished products [33,41,42]. 

Previous studies highlighted breeding, clone selection, genetic engineering, pruning, cluster thinning, deficit irrigation [43], and ultra-high pressure treatment [14] as effective enhancers of phenolic content in grapes and wines. A widely adopted technique in recent years is the use of elicitors [23,43,44,45,46,47]. Elicitors may be exogenous (pathogenic origin) or endogenous (compounds released by plants through pathogenic activity) [48]. The most widely used elicitors are plant extracts, cell wall fragments, avirulent pathogens, and phytohormones (such as methyl salicylate, ethylene, chitosan, benzothiadiazole (BTH), jasmonic acid methyl ester (MeJ), benzoic acid, indoleacetic acid, abscisic acid (ABA), jasmonic acid, salicylic acid (SA), etc.) [4,18,22].

Among elicitors, benzothiadiazole (ASM or BTH, Bion) (Figure 1) has attracted much interest from researchers to improve plants’ resistance to pathogens and enhance the phenolic compounds of grapes [23,33,49]. BTH is an efficient systemic acquired resistance (SAR) booster that has anti-microbial properties and intensifies resistance in the plant by inducing pathogenesis-related protein genes (PR), which activates SAR signal transduction routes, resulting in SAR formation [10,16,18,49,50]. Additionally, BTH treatment can promote the buildup of antifungal compounds such as phenols [22,51] and can be rapidly degraded in plant tissues without producing residues [52].

BTH is known to enhance grape phenolic compounds, which boost resistance to gray mold [10] and delay grapes’ maturation [47,53]. Furthermore, BTH has been found to trigger key enzymes of the secondary metabolism pathway in fruits [10,52,54,55,56]. At the same time, it also activates metabolic pathways [20], causing the biosynthesis of bioactive secondary metabolites [23,44], including phenols and aromatic compounds (Figure 2) [16,33,49,52]. The only side effect of elicitor application is fitness costs which occur preferentially in certain crops but not all, depending on the plant system [57,58]. According to Fumagalli et al. [59], grapevines BTH-treated in growth had no adverse effects on viticultural parameters, yield per vine, or berry size.

Regarding the literature on BTH application, several studies in recent decades have applied BTH to grapevines to improve grape quality by altering the phenolic compounds [2,10,23,30,33,43,44,49,50,60,61]. As a result, we summarize the important findings on the impact of BTH on grape phenolic compounds and highlight some unidentified or understudied biological pathways so that subsequent studies may be geared toward them. 

## 2. BTH Treatment Promotes the Cumulation of Phenolic Substances in Grapes

### 2.1. The Composition of Grape Phenols

Phenolic compounds have peculiar characteristics and are crucial to the metabolism and quality of grapes [62]. The phenolic composition considerably influences the taste, color, aroma, astringency, aging behavior, antioxidant activity, health qualities, and end-product quality of grape berries [40,63,64]. 

In grape research, phenolic compounds are generally classified as either flavonoids or non-flavonoids. Flavonoids consist of a chain of 15 interconnected carbon atoms and two aromatic rings linked by a C6-C3-C6 chain [65,66]. They mainly include chalcones, dihydrochalcones, flavan-3-ols, flavanones, anthocyanins, flavones, and flavonols [67]. On the other hand, non-flavonoids comprise volatile phenols, phenolic acids, stilbenes, and other compounds such as coumarins and lignans [68]. Phenolic acids include hydroxycinnamic acids with nine carbon atoms (C6-C3) and hydroxybenzoic acids with seven carbon atoms (C6-C1) [40]. In addition, studies on grape phenolics often use the term “tannin” to refer to a vast group of polyphenols with various chemical structures and medium to high molecular weights. Condensed and hydrolyzable tannins are the two main classifications of plant tannins [69].

Among the phenolic compounds of *Vitis vinifera* L., anthocyanins (main color pigments) contribute to the color indices of red wines; tannins account for the lasting stability of red wine color. Other phenolic compounds such as phenolic acids, flavanols, and flavonols which act as copigments, affect the color of the wine. These phenols would also affect the aroma [70] and taste of the wine (bitterness and astringency) [71]. Several factors, including grape variety [72], grape maturity [73], changes in predation and disease load, disparities in water-nutrient availability, and illumination and temperature [74,75] can greatly affect the phenolic profile of grapes and their resultant end products. Indeed, some phenolic compounds are induced by stress factors involved in defense mechanisms [43,76]. 

Grape seed, skin, and pulp contain phenolic compounds (Figure 3), but the different fractions contain different amounts [27,77] and change with berry growth and development with spatial and temporal specificity. The phenolic concentration and composition of grape seeds, skins, and pulp vary depending on grape cultivars, soil composition, climatic conditions, geographic origin [27,42,63], cultivation practices [78], or exposure to diseases [79]. Compared to the grape skin and pulp, the seeds have the maximum concentration of phenolic compounds [27,80]; approximately 60–70% of total phenols are preserved in the seeds of grapes [42]. In certain grape varieties, the seeds’ total phenolic content exceeds the skins by more than ten times [80]. The phenolic composition and distribution in grape varieties and cultivars vary [42]. White grapes lack anthocyanins and usually have lower amounts of phenolic compounds than red grapes [77]. The composition and content of phenols in wine are closely related to those in the grapes used for wine production. 

### 2.2. Effect of BTH on the Composition of Grape Phenols

When the grapevine is attacked or under stress, the defenses are activated. The grapevine uses multiple signaling pathways to synthesize defensive compounds based on the kind of attack or stress [81]. One of the resistance processes is known as SAR, which depends on the activation of secondary metabolic pathways and the synthesis of secondary metabolites, including phenolic compounds, in response to an attack or stress on the vine [10,16,42,60].

Without an attack or stress, physical or chemical elicitors may induce defense mechanisms. The formation and buildup of phenolic compounds is a self-defense strategy of grapevine against fungal diseases [42]. Elicitors can activate metabolic pathways of phenols related to defense mechanisms (Figure 2) [22,30,43]. Hence, they are being used as an innovative approach to enhance the phenolic content of grapes as pesticide substitutes to promote plant pathogen resistance [60]. 

BTH is a SA analog that can imitate the function of SA signaling molecules and induce an innate immune response in grapevines [16,22]. Studies have demonstrated that using BTH improves grapes and wines’ phenolic composition and color quality [16,30,44]. In particular, studies on the treatment of various grape varieties with BTH including Merlot [44,52], Monastrell [33,43,44,47,82], Cabernet Sauvignon [47,82], Groppello [2], Syrah [44,61], Cabernet Gernischt [16] and Beauty Seedless [83] as well as their corresponding wines revealed an increase in the concentration of phenolic compounds. However, a study by Paladines-Quezada et al. [47] revealed that BTH decreased the content of phenols in the cell walls of Merlot grapes.

#### 2.2.1. Effect on Anthocyanins

It is well known that plants produce anthocyanins via the flavonoid biosynthetic pathway (Figure 4) [84]. The biosynthesis of anthocyanins can be influenced by BTH and BTH/MeJ blend [43,44,61]. Some studies have shown that applying BTH to grapevines can increase the content of anthocyanins in the grapes [33,44,61].

In Merlot variety, studies showed that anthocyanin concentrations increased in BTH-treated grapes [10,23,52,59,61], causing a slight increase in the resultant wines [59,61]. Gil-Muñoz et al. [61] also found an improvement in the content of most anthocyanins in Merlot grapes treated with BTH except malvidin-3-acetate and coumarate and acetates in the resultant wine. In contrast, a study by Iriti et al. [52] showed a greater increase in malvidin 3-glucoside, malvidin 3-(6-O-acetyl) glucoside, and malvidin 3-(6-O-p-coumaroyl) glucoside levels, which doubled after multiple pre-harvest treatments of grapevines with BTH. Similarly, Fumagalli et al. [59] found that pre-harvest BTH treatments greatly augmented the levels of grape skin anthocyanins, especially monoglucosides of anthocyanidins (cyanidin, malvidin, peonidin, delphinidin, and petunidin) and the corresponding *p*-coumaroyl and acetylated derivatives.

Regarding the Syrah variety, BTH treatments increased the anthocyanin concentration of grapes and corresponding wines, with the total of 3-monoglucosides increasing up to 27% and 10%, except for cyanidin-3-glucoside, in the treated grapes. For the acetylated compounds, BTH increased the contents of petunidin-3-acetyl-glucoside, cyanidin-3-acetyl- glucoside, and peonidin-3-acetyl-glucoside in treated grapes and increased the concentration of cyanidin-3-acetylglucoside and peonidin-3-acetyl-glucoside in the resultant wines. In addition, the concentration of coumarates (peonidin-3-coumaryl-glucoside and cyanidin-3-coumaryl-glucoside) increased in wines produced from the grapes treated with BTH [61].

For the Monastrell variety, some authors found that the anthocyanins content of BTH-treated grapes at harvest was high [23,33,44], but the differences between treatments and the control were more evident at mid-ripening [23,33,46]. Additionally, the rise in the grapes’ anthocyanin content was not observed in the corresponding wines, which could be attributed to the strengthening of the skin cell wall by BTH spraying [30,33]. However, Gil-Muñoz et al. [61] observed lower levels of anthocyanins in BTH-treated grapes and higher levels (up to 10% increase) in the corresponding wines. For individual anthocyanins in BTH-treated grapes, the content of most of them increased except for the tri-OH forms (delphinidin, petunidin, and malvidin), whereas in the corresponding wines, higher acetylated and glucosides anthocyanins were observed compared with the control samples. Furthermore, Ruiz-García et al. [44] found a substantial rise in the anthocyanin content in BTH+MeJ treated Monastrell grapes and the resulting wines. 

#### 2.2.2. Effect on Flavonols

Flavonols are a group of the most studied phenolic compounds for their antioxidant capacity and biological activities [85], as well as their contribution to the color (through co-pigmentation), bitterness, and mouth-feel characteristics of grapes and the corresponding wines [23]. They are products generated from the flavonoid biosynthetic pathway [86], similar to the anthocyanin biosynthetic pathway (Figure 4). Since they share similar pathways, any alteration in enzyme activity associated with the flavonoid biosynthesis pathway would also impact the concentration of flavonols [44]. 

The total content and pattern of flavonols are affected to some extent by BTH [62]. When examining the flavonol compounds of grapes treated with BTH, Gómez-Plaza et al. [23] observed that Syrah grapes treated with BTH had a higher concentration of flavonols from 3 weeks after veraison until the end of maturation, with the highest flavonol concentration at harvest. Similar results were reported by Gil-Muñoz et al. [61], where BTH treatment increased total flavonols by up to 36% with no difference compared to the corresponding wine. However, substantial variations were detected in all individual flavonols analyzed in BTH-treated grapes, except for kaempferol-3-galactoside compared with the control. In wines, quercetin-3-glucoside and kaempferol-3-galactoside levels in BTH-treated samples showed significant differences.

Gómez-Plaza et al. [23] detected a higher concentration of flavonols in BTH-treated Merlot grapes at mid-ripening (two weeks before harvest). Meanwhile, Gil-Muñoz et al. [61] observed that BTH-treated grapes had lower total flavonols content at harvest. However, compared to the control, the wines produced from BTH-treated grapes exhibited greater levels of total flavonols, with increases in all flavonol compounds except isorhamnetin and kaempferols (3-galactoside and 3-glucoside).

In BTH-treated Monastrell grapes, the flavonol content was higher than the control throughout the ripening phase. The greatest differences were observed in control grapes a week before harvest [23]. Contrarily, Gil-Muñoz et al. [61] found no significant difference in the total flavonol content in treated grapes compared with the control. Nevertheless, variations were observed in individual flavonols between treated and untreated grapes at harvest, including isorhamnetin-3-glucoside with a higher concentration. In the corresponding wines, statistical differences were found for kaempferol-3-glucoside, galactoside, and syringetin-3-glucoside. Ruiz-García et al. [44,87] also found that applying BTH on grapes from Monastrell clones increased the biosynthesis of flavonols and improved the flavonol content of the grapes at harvest. 

Nonetheless, in two years of investigation, the results were not entirely consistent. Similarly, Ruiz-García et al. [88] noticed in a two-year experiment that the flavonol concentration in wine made from BTH-treated Monastrell had slight variations compared to the control wine, with significant differences only in one year. Comparable results were also found in other studies [33,46]. The results of these studies suggest that the effect of BTH treatment on the flavonol composition of grapes and the corresponding wines varies depending on the year of production.

#### 2.2.3. Effect on Proanthocyanidins (Tannins)

There are two kinds of tannins in wine: condensed tannins from grapes and hydrolyzable tannins from oak barrels. In this review, more focus is placed on condensed tannins from grapes. In general, condensed tannins are also referred to as proanthocyanidins. 

The application of BTH can increase the proanthocyanidin content of Merlot grapes by up to 36% and reduce the incidence and consequences of gray mold [10]. Ruiz-García et al. [88] and Gómez-Plaza et al. [23] reported that BTH application increased the proanthocyanidin content of Monastrell grape skins at harvest. Likewise, Ruiz-García et al. [43] discovered that BTH treatment elevated the content of proanthocyanidins in the skin of Monastrell grape clones. A study by Ruiz-García et al. [88] showed higher mDP values of proanthocyanidins in BTH-treated grapes. According to the study, more polymerized proanthocyanidins inhibited the activity of fungal enzymes greater than less polymerized proanthocyanidins [89]. The authors explained that the higher level of proanthocyanidins and their higher mDP values due to BTH application is part of the grape resistance mechanisms induced by BTH [10,88]. In contrast, the authors observed a lower concentration of proanthocyanidins in Syrah and Merlot grapes treated with BTH than in control grapes at harvest [23,61,88]. However, during ripening, Monastrell, Syrah, and Merlot grapes treated with BTH had higher levels of proanthocyanidins than control grapes a week before harvest [23,61]. In Merlot, BTH-treated grapes showed higher levels of proanthocyanidins in the skin than control grapes, which persisted until a week before harvest and then decreased to lower levels than control grapes [61,88].

In addition, the previous study found that BTH treatment showed partial differences in promoting proanthocyanidin synthesis in grapes. Gil-Muñoz et al. [61] reported that BTH treatment affected the mDP of grape seeds, skins, and resulting wines and increased the percentage of galloylation in treated grapes’ skins and wines. In Syrah, the BTH-treated samples showed differences in tannin (proanthocyanidin) composition compared to the control grapes, which were more noticeable in the grape skins and resultant wines than in the grape seeds. Similar results were also observed in Merlot and Monastrell grapes [61]. 

Furthermore, significant differences were observed in the terminal and extension units of skin tannins in the control and BTH treatments. In particular, only (+)-catechin (terminal unit) showed significant variations, while the content of (+)-catechin, (−)-epicatechin, and epicatechin-3-o-gallate (terminal units) increased in the control samples compared to the BTH treatments. In seeds, the content of (+)-catechin and (−)-epicatechin increased in BTH treatments [61]. In the Merlot variety, the terminal units only increased the percentage of (+)-catechin and epicatechin-3-O-gallate in the skins and (+)-catechin in the seeds after BTH treatment. The extension units showed an increase in the seeds’ epicatechin-3-O-gallate and skins and wines’ epigallocatechin [61].

#### 2.2.4. Effect on Stilbenes

In recent years, multiple pre-harvest BTH treatments have been shown to promote the buildup of *cis-* and *trans*-resveratrol in the skin of Merlot grapes with the increase in *trans*-resveratrol being significantly higher than that of *cis*-resveratrol [52]. Similarly, BTH treatment may considerably increase *trans*-resveratrol concentration in Syrah grapes [53] and stilbene concentration in Monastrell grapes [87]. This may be associated with a delay in ripening [53]. The development of stilbenes is probably related to the mechanism by which BTH promotes a defense reaction in grapes. For example, in grape cells treated with 0.1 or 1 mmol/L^−1^ BTH, a significantly higher amount of stilbenes was found compared to control grapes during incubation [22]. 

#### 2.2.5. Effect on Color Quality

Based on the report by Beckers and Spoel [81], the increase in color intensity of grapes is a response to the increase in phenolic compounds. The improvement of color quality of grapes and wine by BTH treatment has been reported in some studies [16,47,61,88]. In one study, higher CIRG values were measured in Cabernet Gernischt grapes treated with BTH than in control grapes [16]. Similarly, BTH treatment increased the chromatic characteristics of Syrah and Merlot grapes and their corresponding wines [49,61]. Wines made from BTH-treated Monastrell [44,47,61,88] and Merlot and Syrah [62] grapes also showed higher color intensity, while the tone and lightness were barely affected by the BTH treatments [23]. 

Altogether, BTH can affect the buildup of phenolic compounds in several ways: directly (by altering the biosynthesis of the molecules) and indirectly (by altering the molecular content due to differences in the volume and weight of the fruits). It has been shown that the effects of BTH treatments on grape phenolic content strongly depend on the variety [61,88], climate [47], environmental conditions [30], timing of BTH treatments [23,33,90], and maturity [23]. The highest content of phenolic compounds is not always achieved at maturation but a few days before harvest [23]. The years with higher humidity and lower temperatures could offer suitable conditions for pathogen growth and contribute to a more efficient response of plants treated with BTH, especially by activating the phenylpropanoid pathway [61,88].

Regarding BTH treatment time, Boss et al. [90] reported that it should be applied in a week during veraison which is the crucial phase of phenols development, whereas Gómez-Plaza et al. [23] suggested it should be applied as close to the harvest date as possible to obtain the most significant effect. Paladines-Quezada et al. [33], on the other hand, have shown that the mid-ripening phase is the most appropriate for treating Monastrell grapes with BTH to achieve maximum phenolic accumulation at harvest. 

In wines made from BTH-treated grapes, the increase in anthocyanin content is not usually observed [33]. Due to variations in skin cell wall components, there have been discussions on the challenge of extracting phenolic compounds from grapes to wine [91,92]. BTH treatment has been reported to increase cellulose concentration in skin cell walls at veraison and mid-ripening [33]. Moreover, BTH applications can cause changes to the skin cell wall structure and composition of Cabernet Sauvignon, Merlot, and Monastrell grapes [82]. Therefore, the application of BTH affects the cell walls of berry skins, which can consequently affect the anthocyanin content of corresponding wines.

## 3. Effect of BTH on Phenols Metabolism

### 3.1. Synthesis of Plant Phenols

The soluble form of phenolics is mainly located in the vacuoles of plant cells, which may be in free or conjugated form, while the insoluble phenolics are mainly found in the cell wall matrix [27,93]. The veraison is the critical period in grapes during which phenols develop [52]. Phenolic compounds are usually synthesized from phenylalanine or tyrosine through the shikimic acid pathway in plant intracellular organs during the growth of plants [27,64,66]. They consist of an aromatic ring with one or more hydroxyl substituents and range from monomeric molecules to highly polymerized compounds [66,94]. The hydroxyl substituents on the aromatic ring are responsible for the antioxidant properties of the phenolic compound [66]. 

The synthesis of phenolic compounds originates from the branching of phenylpropanoids (Figure 4) [44,62,95]. The first step of synthesis begins with the deamination of phenylalanine catalyzed by phenylalanine ammonia-lyase (PAL) [61]. Phenylalanine is a product of the shikimate pathway, which links carbohydrate metabolism with the production of aromatic amino acids and secondary metabolites [62]. In the second step, phenol skeletons are derived from malonyl-CoA and *p*-coumaroyl-CoA, which are biogenetically derived from phenylpropanoids and acetate pathways [62]. Subsequently, malonyl-CoA and *p*-coumaryl-CoA are transformed into phenols by stilbene synthase (STS) and chalcone synthase (CHS) through the formation of an aromatic ring (by adding three more carbon groups consisting of two C atoms) [44,52]. Through a bifurcation of this pathway, two major classes of phenolic compounds, flavonoids (by CHS) and stilbenes (by STS), can be synthesized. In addition, the flavonoid pathway leads to the synthesis of flavan-3-ols, flavonols, anthocyanins, and proanthocyanidins [62,96,97]. 

### 3.2. Influence of BTH

Presently, there are more studies on pre-harvest BTH treatment to enhance phenolic accumulation in grapes but fewer studies on the effects of BTH treatment on the phenolic metabolism of grapes. Based on the mechanism of SAR pathway, BTH (functional analog of SA) can stimulate the expression of protein defense genes (PR1, PR2, and PR5) associated with pathogenesis via the protein non-expressor gene 1 (NPR1), which is also associated with pathogenesis [48,98,99]. Expression of the protein defense genes then activates the SA signaling pathway leading to SAR establishment [61,98] and encoding protein defense and key enzymes of secondary metabolism (Figure 5) [52,59,100]. This is also accountable for the increase of phytoalexins, synthesis of protein defense genes, and reinforcement of cell walls, among others [18]. It has been reported that BTH inhibits ethylene and malondialdehyde (MDA) [101] but enhances the catalytic activity of various enzymes, particularly PAL, thereby influencing the synthesis of secondary metabolites in plants [10,22,50,102,103,104]. Paladines-Quezada et al. [30] suggested that grapevine cells treated with elicitors such as BTH activate defense responses that produce high amounts of superoxide radicals, H_2_O_2_, and other reactive oxygen species (ROS) in the cell wall. ROS can cause rapid cross-linking or interlacing of cell wall phenolic compounds. Wang et al. [22] also found that BTH can activate a SAR defense response in grape suspension cells and enhance the accumulation of stilbene phytoalexins, VvNPR1.1 and PR1 genes expression, and the cellular burst of H_2_O_2_. Cellular hypersensitive defense responses, including activation of defense genes and induction of defense compounds (e.g., phenols) in plants during oxidative burst could be attributed to the accumulation of H_2_O_2_ in plant cells [105]. Using elicitors such as BTH can induce early and rapid H_2_O_2_ production in cultured *V. vinifera* cells, triggering the expression of defense-related genes or phytoalexins (e.g., phenols) synthesis [22,106,107].

However, BTH can reduce the levels of primary metabolites such as amino acids and soluble sugars in treated grapes [16,22]. In particular, total soluble sugar and soluble sugar composition may vary significantly in BTH-treated berries [22]. Soluble sugars are not only responsible for fruit organoleptic quality but also act as key signaling molecules that can modulate the transcription of genes involved in defense responses and metabolic processes, thus affecting the biosynthesis of secondary metabolites [22,108]. Hence, there are soluble sugars associated with BTH effect on the phenolic compounds in grapes and the defense mechanisms induced by BTH. BTH treatment can stimulate lower sucrose-synthesizing enzymes (SS-synthesis, SPS, and SPP and higher sucrose-hydrolyzing enzyme (SS-cleavage), triggering a slow increase in sucrose breakdown, a decrease in glucose level and the buildup of fructose in grapes [22]. In addition, sugar accumulation in grape pulp and skin can also enhance the synthesis of anthocyanins [52,109,110]. Therefore, soluble sugars are associated with the impact of BTH on the phenolic contents of grapes and the defense mechanisms induced by BTH. Some authors [22] suggest that the phenylpropanoid pathway and the common precursor of sucrose metabolism (UDP-glucose) may be directed toward the biosynthesis of phenolic compounds by BTH treatment, whereas soluble sugar accumulation might reduce. 

The increase of anthocyanins and stilbenes content in BTH-treated grapes could also be due to BTH induction of PAL [58]. Iriti et al. [52] found that the buildup of stilbenes in the berry skin decreased during the ripening period of untreated berries while the production of anthocyanins increased, possibly due to competition between the two branches of the phenylpropanoid pathway and the different regulation of the key enzymes STS and CHS (Figure 4). Meanwhile, it appears that BTH treatment reverses the opposite association between anthocyanin and resveratrol pathways to some extent. Hence, BTH may reduce competition between STS and CHS, enabling substrate binding and increasing anthocyanin and resveratrol synthesis. In BTH-treated berries, anthocyanin accumulation does not necessarily affect resveratrol synthesis during ripening. In other words, the usual metabolic switch between the two branches of the same metabolic pathway seems to be avoided by BTH treatment [52].

Anthocyanins can be affected by BTH treatments [23,111]. Gómez-Plaza et al. [23] demonstrated that BTH could activate enzymes related to anthocyanin metabolism. Repka et al. [112] and Gozzo [113] also reported that BTH treatment promoted the activities of enzymes (i.e., chalcone isomerase and PAL) in the phenylpropanoid pathway. Furthermore, BTH treatment can contribute to stilbenes and flavan-3-ols synthesis due to SAR and BTH induction of the expression of phenylpropanoid genes in grapevine [62]. In addition, in light of the effect of BTH treatment on grape polyphenols and the findings of earlier studies, Fumagalli et al. [59] suggested that BTH could enhance the activity of CHS during polyphenol or anthocyanin biosynthesis.

Studies on pre- or post-harvest BTH treatment have been conducted not so much in terms of secondary metabolic mechanisms in grapes as in various plants. In one study, post-harvest BTH treatment resulted in a higher anthocyanin content in strawberries and increased the activities of metabolic enzymes such as PAL, 4-coumarate/coenzyme A ligase (4-CL), cinnamate-4-hydroxylase (C4H), dihydroflavonol 4-reductase (DFR), glucose-6-phosphate dehydrogenase (G6PDH), tyrosine ammonia-lyase (TAL), and shikimate dehydrogenase (SKDH) [111]. This suggests that the increase in anthocyanin content by BTH may be due to the activation of associated metabolic enzymes. In addition, post-harvest BTH treatment resulted in the activation of PAL in peaches and mangoes [114,115], while the activity of peroxidases (POD) and polyphenol oxidase (PPO) increased in mangoes [115]. Similarly, post-harvest BTH treatments also induced PPO and POD activity in bananas [116] and loquats [117]. Interestingly, this approach has been reported to increase the total phenolic compounds in mangoes and bananas [115,116]. Furthermore, the exogenous application of BTH increases the expression of proanthocyanidin-related MBW complex (MYB-bHLH-WD40) and the content of flavan-3-ol in general [118], which promotes the accumulation of proanthocyanidins. Similar results were also reported by Felicijan et al. [119]. However, the activity and gene expression of phospholipase (phospholipase A2PLA2, phospholipase CPLC, and phospholipase DPLD) were inhibited by post-harvest BTH treatment in melons [54].

Overall, the influence of BTH on grapes has been reported in the literature, particularly on grape color, phenolics, variety, and metabolism (Table 1). Based on the existing literature, many studies have been conducted on the effects of BTH on grape phenolics in relation to variety, but less on the effects on grape metabolism and sensory characteristic such as color and mouthfeel (which are associated with phenolic compounds). Hence, future studies can be conducted in these areas.

## 4. Conclusions and Future Perspectives

BTH treatment is a good strategy to improve the synthesis of grape phenolic compounds. However, BTH treatments do not always produce consistent results. The effects of BTH treatment rely on several factors, including variety, climate, environmental conditions, the timing of BTH treatments, and maturity. This review found that BTH can improve phenolic compounds and color quality by directly affecting secondary metabolism or altering primary metabolism. Although some encouraging progress has been made, the complete process is not yet clearly understood. The influence of BTH on phenolic compounds and their synthesis is still unclear. In particular, the metabolic and physiological changes resulting from the inducer are still unknown and require further studies. In the future, a clear understanding of the mechanisms of BTH at the cellular, molecular, and genetic levels would help overcome these problems. In addition, focusing on the influence of climate change, variety, and environmental conditions on BTH-induced phenols accumulation in grapes can help producers use BTH scientifically, effectively, and safely. Therefore, the comprehensive and in-depth analysis of the regulatory mechanism of phenols enhancement by BTH is of great importance for improving our knowledge of the regulatory control of phenol synthesis in grapes.

## Figures and Tables

**Figure 1 foods-11-03345-f001:**
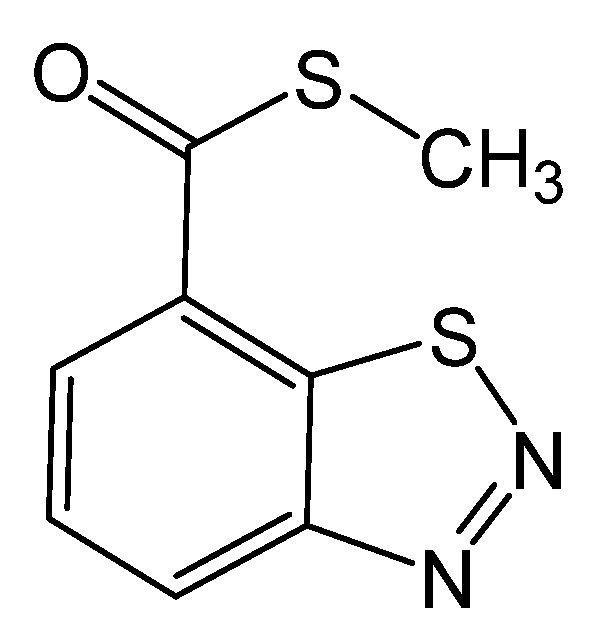
Chemical structure of benzothiadiazole.

**Figure 2 foods-11-03345-f002:**
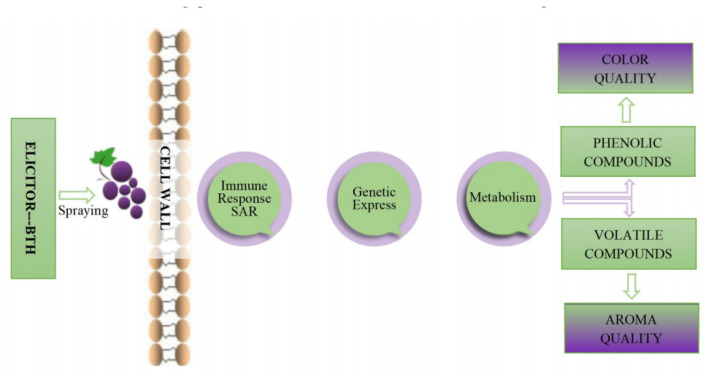
Impact of BTH spraying on the color and aroma of grapes.

**Figure 3 foods-11-03345-f003:**
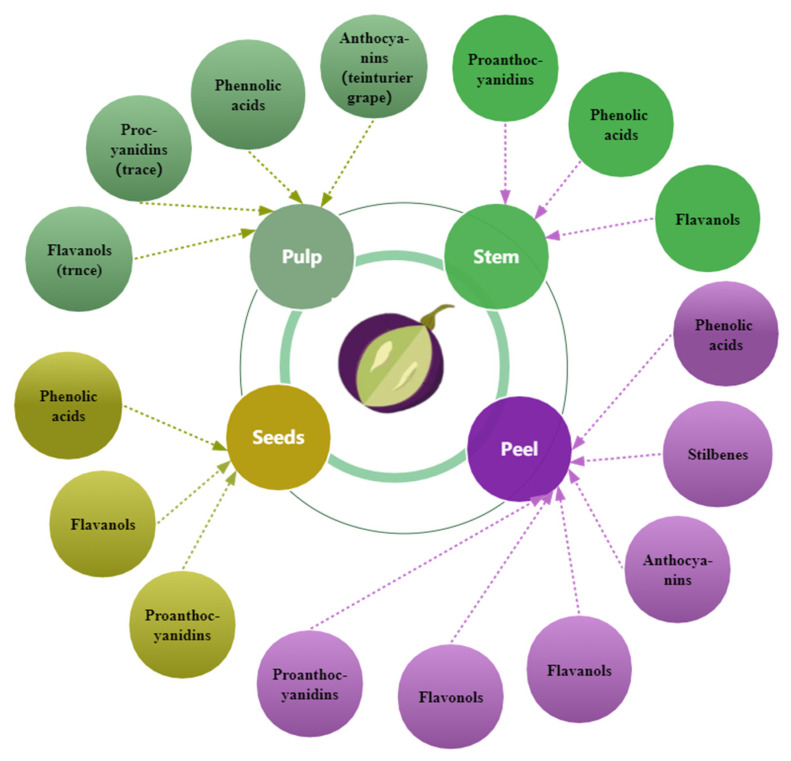
The primary distribution of phenols in grapes.

**Figure 4 foods-11-03345-f004:**
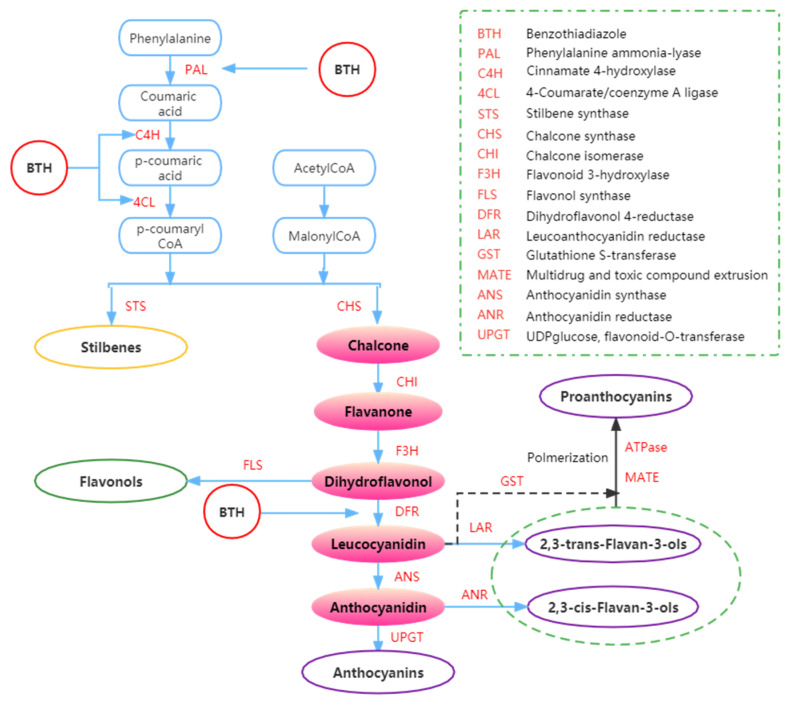
Summary of the effects of BTH on phenols metabolism. Adapted from Ruiz-García and Gómez-Plaza [43].

**Figure 5 foods-11-03345-f005:**
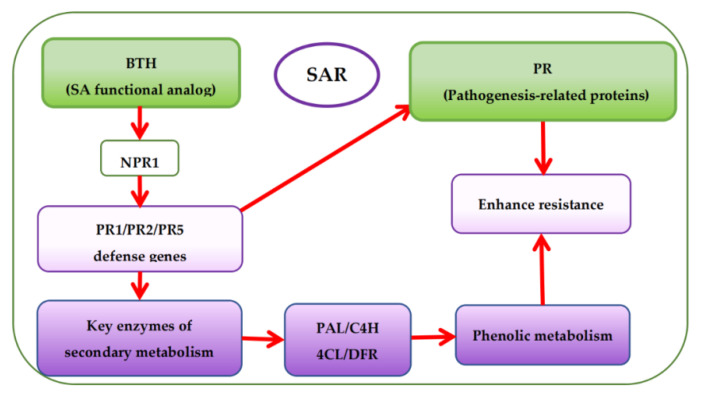
Overview of systemic acquired resistance (SAR) and phenolic accumulation mechanism with BTH treatment.

**Table 1 foods-11-03345-t001:** Summary of literature on BTH impact on grape variety, phenols, color, and metabolism.

Research Area	Subclasses	Reference
Variety	Monastrell	[23,30,33,44,46,47,61,82,87,88]
	Merlot	[10,23,47,52,61,88]
	Syrah	[23,49,53,61,88]
	Cabernet Sauvignon	[47,82]
	Groppello	[2]
	Beauty Seedless	[83]
	Cabernet Gernischt	[16]
Phenols	Anthocyanin	[10,23,30,33,44,46,52,61]
	Flavonols	[23,33,44,46,87,88]
	Proanthocyanidins (Tannins)	[10,23,50,61,88]
	Stilbenes	[22,52,53,87]
Metabolism	Enzymes	[10,22,23,50,52]
	Hydrogen dioxide and ROS	[23,30]
Color	Color intensity	[16,23,44,47,61,88]

## Data Availability

Data is contained within the article.

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
