# Peer review of "Pre-Harvest Benzothiadiazole Spraying Promotes the Cumulation of Phenolic Compounds in Grapes"

_foods, 2022, doi:10.3390/foods11213345_

Round 1

Reviewer 1 Report

Dear Authors,

The manuscript needs minor revisions. All suggestions are in comments in the manuscript.

Reviewer 2 Report

The review deals mainly polyphenols and only in the last part speaks about the potential effect of BHT on polyphenol metabolism, reporting results of some papers. I did not find in the reference any of papers on BHT done by the authors unless the ones published on POSTEC at name W. Jiang is the same author Yumei Jiang of this review.

I expect that Authors would have done some work on polyphenol biosythesis as affected by BHT but apparently they didn't

Reviewer 3 Report

Manuscript ID: foods-1900225

Title: Pre-harvest BTH Spraying Promotes the Cumulation of Phenolic Compounds in Grapes

This manuscript provides a review on the use of benzothiadiazole (BHT) sprayed in pre-harvest to promote the accumulation of phenolic compounds in grapes. The work is well written and interesting for the part concerning the effect of BTH on phenolic metabolism, which has been poorly studied. In my opinion, the manuscript needs some revisions.

ABSTRACT

I suggest to better formulate and state the aims of the work. In particular, avoid redundancy and define the scope only once. As it is now the abstract, two distinct aims are reported at lines 13-14 and 16-17.

I think that more efforts should be done by the authors to better present the existing literature on the topic and to critical evaluate it. Practically could be very useful to the readers to add in the paper table(s) and/or figure(s) that easily present the literature classified on the base of, for example, country, grape variety, compound studied, etc. (all significant info). In this way, the paper could furnish an easy access to the framework from which, according to the authors’ aim, critical steps not yet well investigated could be easy highlighted. Pie charts for example could also be useful to easily draw info about the amount of papers produces on one variety, in one country, and so on.

Lines 32-33: the sentence seems not complete: “Chemical pesticides are the most effective and common method” for what?

Sub-section 3.5 Effect on Stilbenes

An overview of the effect of BTH treatment on stilbenes is only given in lines 412-420. Instead, in lines 421-445 of the same sub-section, the effect of BTH treatment on other phenolic compounds is reported. Therefore, the authors should extend the overview of BTH treatment on stilbenes from the literature. Furthermore, the authors should move the effects of BTH on other phenolic compounds to the most appropriate sections. I suggest moving lines 421-436 after line 259 and moving lines 437-445 to the section on the effects of BTH on anthocyanins.

Finally, what also is missed is a paragraph providing a critical evaluation of the state of the art on the topic, that summarize the main findings and highlights the lacks for further research.

Minor mistakes

Please check carefully typing errors such as in line 279 “and acetates” that is duplicate…….

Round 2

Reviewer 2 Report

The Authors confirmed that they never made research in this field thus they did not publish anything. This is not what we intend to carry out a review in a science.

Reviewer 3 Report

The problems have all been solved.

Author Response

Thank you very much for your review and recommendation!